# DiffSound: Differentiable Modal Sound Simulation for Inverse Reasoning

## Abstract

Accurately estimating and simulating the physical properties of objects from real-world audio observations is of great practical importance in the fields of vision, graphics, and robotics. However, previous differentiable rigid or soft body simulations cannot be directly applied to modal sound synthesis due to the high sampling rate of sound, and previous audio synthesizers do not fully model the physical properties of objects behind the modal analysis. We propose DiffSound, a differentiable sound simulation framework for physically based modal sound synthesis. Our framework can solve a wide range of inverse problems due to the differentiability of the entire pipeline, including a variety of object's properties embedded and their gradients backpropagation. Experimental results demonstrate the effectiveness of our approach, highlighting its ability to reproduce the target sound accurately and reason the physical parameters such as material, geometry shape, and impact position. Our differentiable sound simulator serves as a valuable tool for applications requiring sound synthesis and analysis.

## 1 Introduction

The concept of differentiable simulation has become increasingly popular in the graphics and machine learning communities in recent years (Popović et al., 2003; de Avila Belbute-Peres et al., 2018; Toussaint et al., 2019; Degrave et al., 2019; Qiao et al., 2020; Xu et al., 2021). A differentiable simulation framework allows for gradient-based optimization and can be integrated into a neural network for end-to-end learning.

Our work focuses on differentiable sound simulation, which addresses a unique challenge compared to standard differentiable rigid or soft body simulations (Hu et al., 2020; Geilinger et al., 2020; Du et al., 2021; Degrave et al., 2019; Qiao et al., 2020; Xu et al., 2021) due to the high sampling rate of sound. While previous audio synthesizers (Engel et al., 2020; Clarke et al., 2021) can optimize for many audio and physical-based properties, they are unable to explicitly model more fundamental physical properties such as Young's modulus, Poisson's ratio, size or shape, and impact position, which are critical for realistic modal sound synthesis.

Inferring these objects' properties from real sound recordings can potentially enable various Real-to-Sim applications. For example, we can accurately infer material parameters from real-world recordings and use them to create realistic virtual objects, such as those in Gao et al. (2021; 2022; 2023). We can also leverage a differentiable sound simulation framework to design the shape and material of virtual objects to produce the desired sound, and then transfer the results back to real objects using 3D printing technology (Bharaj et al., 2015). The information about an object's shape, material, and impact position can also complement visual perception, particularly in cases of low visual resolution or poor lighting, for multisensory robotic applications (Clarke et al., 2021; Li et al., 2022a).

Towards this end, we introduce DiffSound, a differentiable simulator for physically-based modal sound synthesis, which employs a high-order finite element method to model the physical properties of objects and establish a seamless, fully differentiable connections between the recorded audio and these physical properties.

Our DiffSound differentiable sound simulation framework consists of three main components. First, we propose a differentiable shape representation that combines implicit neural representation

and explicit 3D tetrahedral mesh representations for sound simulation. Second, we introduce a high-order finite element analysis module that allows for incorporating differentiable material and shape parameters. Finally, we design a differentiable audio synthesizer with a hybrid loss strategy to enable smooth optimization of the entire differentiable simulation framework.

We demonstrate the effectiveness of our differentiable sound simulation framework through a wide range of inverse problems, including physical parameter estimation, impact position estimation, and object shape estimation, from synthetic data or real sound recordings.

## 2 RELATED WORK

Our work is closely related to the simulation of modal sounds and its applications and high-order FEM in computer graphics. It is also relevant to the recently developed differentiable simulation methods in the graphics and machine learning communities.

**Modal Sound Synthesis** Modal sound synthesis is a technique that has been used to synthesize sounds of rigid bodies (van den Doel et al., 2001; O'Brien et al., 2002; Raghuvanshi & Lin, 2006). These methods compute the vibration modes of a 3D object through a generalized eigenvalue decomposition. Based on the basic modal sound method, many complex sound phenomena can be simulated, such as knocking, sliding, and friction sound (van den Doel et al., 2001), acceleration noise (Chadwick et al., 2012), complex damping sound (Sterling et al., 2019), and high-quality contact sound (Zheng & James, 2011).

Our work also relates to previous endeavors focused on estimating material parameters using pre-recorded audio clips (Ren et al., 2013; Zhang et al., 2017). In contrast to earlier approaches, our work offers an end-to-end optimization-based solution to these problems, resulting in enhanced accuracy. Compared with prior methods that optimize object shapes to achieve desired sounds (Bharaj et al., 2015), our approach optimizes all modes of the generated sound, rather than focusing on a single fundamental frequency. Additionally, our approach provides more flexibility in shape optimization, going beyond simple scaling and stretching.

**High-Order FEM** In engineering, higher-order methods are often preferred over lower-order methods due to their superior accuracy and convergence properties. In computer graphics, finite element methods (FEM) with linear shape functions is prevalent due to its simplicity and computational efficiency. While limited prior work demonstrates that higher-order methods have the potential to produce better simulation results (Mezger et al., 2008; Bargteil & Cohen, 2014; Schneider et al., 2019; Longva et al., 2020), they are not commonly used in the field.

To the best of our knowledge, the sole previous attempt at incorporating high-order FEM into modal sound synthesis is documented in (Bharaj et al., 2015), where results from the engineering software COMSOL (COMSOL AB, Stockholm, Sweden, 2005) are employed directly. In contrast, within our differentiable framework, we implement a high-order FEM approach to guarantee both high-quality sound simulation and differentiability.

**Differentiable simulation** Differentiable simulation has recently gained much popularity in the graphics and machine learning communities. Several advances have been made in this field with differentiable simulators designed for rigid-body dynamics (Popović et al., 2003; de Avila Belbute-Peres et al., 2018; Toussaint et al., 2019; Degrave et al., 2019; Qiao et al., 2020; Xu et al., 2021), soft-body dynamics (Hu et al., 2019; Hahn et al., 2019; Hu et al., 2020; Geilinger et al., 2020; Du et al., 2021), fluid dynamics (Treuille et al., 2003; McNamara et al., 2004; Wojtan et al., 2006; Schenck & Fox, 2018; Holl et al., 2020), and cloth (Liang et al., 2019; Murthy et al., 2021; Li et al., 2022b).

There are also differentiable rendering methods proposed for signal processing (Engel et al., 2020) and modeling impact sound (Clarke et al., 2021). These methods can capture various physical-based properties, such as modal response and force profiles. However, they do not explicitly consider the fundamental physical properties of objects, such as shape, material, size, and impact position. Another promising approach uses neural networks to approximate the modal analysis process (Jin et al., 2020; 2022). Although neural networks are inherently differentiable, ensuring physical accuracy can

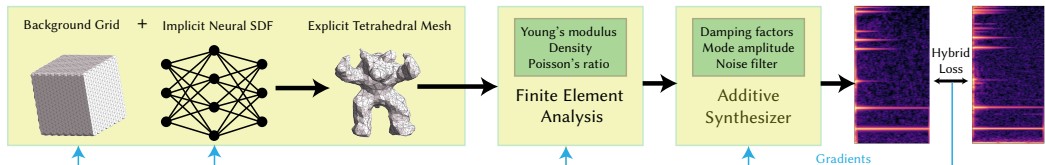

Figure 1: Our DIFFSOUND simulation pipeline. The differentiable tetrahedral mesh representation is employed to directly optimize the topology of a tetrahedral mesh. Subsequently, a differentiable high-order finite element analysis module is utilized to analyze the vibration frequencies of the tetrahedral mesh. Finally, a differentiable additive synthesizer produces the impact sound, and a hybrid loss function optimizes all learnable modules separately or simultaneously.

be challenging and accurate modal analysis may not be achieved just through neural network optimization.

## 3 DIFFERENTIABLE SOUND SIMULATION

This section elucidates the core algorithms of our differentiable sound simulation framework, as illustrated schematically in Fig. 1. Our model hinges on a specialized differentiable tetrahedral mesh for sound simulation, as detailed in Sec. 3.1. Subsequently, in Sec. 3.2, we expound on the differentiable high-order finite element method (FEM) for modal analysis. Finally, we delineate the optimization process's loss function in Sec. 3.3.

### 3.1 DIFFERENTIABLE TETRAHEDRAL REPRESENTATION

We propose a differentiable tetrahedral mesh representation designed for our differentiable simulations, building upon the foundation of Deep Marching Tetrahedra (DMTet) (Shen et al., 2021; Munkberg et al., 2022). Our approach involves the representation of a shape through a Signed Distance Field (SDF) implicitly encoded by a Multilayer Perceptron (MLP) (Sec. 3.1.1), which is then transformed into an explicit tetrahedral mesh using a deformable tetrahedral grid (Sec. 3.1.2).

#### 3.1.1 IMPLICIT NEURAL REPRESENTATION

Given the inherent limitations in precisely associating the sound of an object with its exact shape, there is a potential for significant ambiguity in the resulting geometry when optimized by sound. To tackle this challenge, we utilize a Multilayer Perceptron (MLP) to parameterize the SDF values. This implicit parameterization effectively serves to regularize both the SDF and the overall smoothness of the reconstructed shape. Furthermore, the degree of smoothness can be controlled by adjusting the frequency of the positional encoding proposed in Neural Radiance Fields (Mildenhall et al., 2020), which is applied to the inputs of the MLP.

#### 3.1.2 FROM IMPLICIT TO EXPLICIT REPRESENTATION

We customize the Marching Tetrahedra (MT) (Doi & Koide, 1991) algorithm to convert the encoded Signed Distance Function (SDF) into an explicit tetrahedral mesh. The vertices in the background tetrahedral cells are also deformable within a half-cell size range, allowing for stronger geometric expression capability. By utilizing the SDF values of the vertices within a tetrahedron obtained from the MLP, MT discerns the surface typology within the tetrahedron based on the signs of the SDF values. Our modification focuses on identifying the internal tetrahedron rather than the surface typology, as depicted in Figure 2. This process results in a total of five distinct configurations, accounting for rotation symmetry. The location of surface vertices is computed through linear interpolation along the edges of the tetrahedron, similar to the methodology employed in DMTet (Shen et al., 2021; Munkberg et al., 2022). If the internal subregion is more complex than a tetrahedron, we subdivide it into smaller tetrahedrons. Finally, we extract the largest connected tetrahedral mesh to eliminate high-frequency noise interference from the sound of small fragments during optimization.

## 3.2 DIFFERENTIABLE HIGH-ORDER FEM

Prior studies (Hughes, 2012; Bharaj et al., 2015) have noted the limitations of linear tetrahedral finite elements in producing accurate solutions, even with refined simulation discretization. In this work, we propose the use of differentiable high-order FEM for greater accuracy and generality.

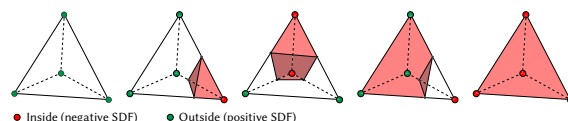

Figure 2: Five configurations of the interface between background tetrahedrons and internal ones. If the internal subregion is more complex than a tetrahedron, it will be subdivided into smaller tetrahedrons.

We compute the mass and stiffness matrices for the tetrahedral mesh (introduced in the above Sec. 3.1) to be differentiable with respect to the material coefficients, namely Young's modulus, density, and Poisson's ratio as introduced in Sec. 3.2.1. Subsequently, in Sec. 3.2.2, we compute the gradient from the eigenvalues obtained through eigendecomposition with respect to these two matrices. For a comprehensive derivation of these matrices, please refer to (Sifakis & Barbic, 2012; Zhu, 2018).

### 3.2.1 MASS AND STIFFNESS MATRIX

To obtain the mass matrix, we initially compute the element matrix for each individual tetrahedral element, followed by the assembly process to construct the mass matrix for the entire tetrahedral mesh. Let $V$ denote the volume occupied by a tetrahedral element, $\rho$ represents its density, and the shape function value at position $x$ with respect to node $i$ is denoted as $N_i(x)$. The element mass matrix $\mathbf{M}_e$ is defined as follows:

$$\mathbf{M}_e^{ij} = \rho \iiint_{x \in V} N_i(x)N_j(x)dx. \tag{1}$$

To compute this volume integral, we employ the Gaussian numerical integration method, selecting $t$ Gaussian integration points $g_k$ within the tetrahedral element, with corresponding Gaussian integration weights $w_k$. The unit mass matrix can be calculated as follows:

$$\mathbf{M}_e^{ij} = \rho V \sum_{k=1}^{t} N_i(g_k)N_j(g_k)w_k. \tag{2}$$

For a high-order tetrahedral element containing $n$ nodes, the algorithm described above yields a unit mass matrix $\mathbf{M}_e$ of size $3n \times 3n$. Now, for the entire tetrahedral mesh with a total of $m$ nodes, it is only necessary to add each element $\mathbf{M}_e^{ij}$ computed for each tetrahedron to the corresponding entries $\mathbf{M}^{ij}$ of the overall mesh's mass matrix $\mathbf{M}$. This assembles a $3m \times 3m$ mass matrix $\mathbf{M}$.

Following the defined process for the mass matrix, let $E$ denote Young's modulus and $\nu$ denote Poisson's ratio. The element stiffness matrix $\mathbf{K}_e$ of size $3n \times 3n$ for a tetrahedral element is defined as follows:

$$\mathbf{K}_e = \sum_{k=0}^{t} w_k V \mathbf{D}(g_k)^T \mathbf{B}(E, \nu) \mathbf{D}(g_k). \tag{3}$$

Here, $\mathbf{B}(E, \nu)$ is the elasticity matrix representing the material model, and we adopt the linear elastic model (Sifakis & Barbic, 2012). $\mathbf{D}(g_k)$ is a matrix derived from the shape functions at point $g_k$. To construct the overall stiffness matrix $\mathbf{K}$ for the entire tetrahedral mesh, we add each element in $\mathbf{K}_e$ computed for each tetrahedron to the corresponding entries of the overall mesh's stiffness matrix $\mathbf{K}$. This assembles a $3m \times 3m$ stiffness matrix $\mathbf{K}$.

We employ PyTorch (Paszke et al., 2017) to efficiently batch calculate both the element mass matrix and element stiffness matrix. Subsequently, these element matrices are assembled into global Coordinate Format (COO) sparse matrices for further processing. Notably, it's essential to highlight that these computations are automatically differentiable, enabled by PyTorch. Additionally, both the mass and stiffness matrices exhibit differentiability with respect to the material properties ($\rho$ in the mass matrix and $\mathbf{B}(E, \nu)$ in the stiffness matrix), as well as the geometry derived from our differentiable tetrahedral mesh ($N_i(x)$ in the mass matrix and $\mathbf{D}(g_k)$ in the stiffness matrix, as well as $V$ in both cases).

### 3.2.2 EIGENVALUE DECOMPOSITION

Now, we perform a generalized eigenvalue decomposition on the mass and stiffness matrices as:

$$\mathbf{KU} = \mathbf{MU\Lambda}, \tag{4}$$

where $\mathbf{U}$ is a stack of $k$ eigenvectors, and $\mathbf{\Lambda}$ is the diagonal matrix of $k$ eigenvalues. The $i$th eigenvector, denoted as $\mathbf{u}_i$, represents the surface vibration distribution of the $i$th mode, while the $i$th eigenvalue, $\lambda_i$, determines its frequency and satisfies $\mathbf{Ku}_i = \lambda_i \mathbf{Mu}_i$.

Taking the derivative of both sides with respect to $\lambda_i$ in the equation $\mathbf{Ku}_i = \lambda_i \mathbf{Mu}_i$, we obtain:

$$\partial \mathbf{Ku}_i + \mathbf{K}\partial \mathbf{u}_i = \lambda_i \mathbf{M}\partial \mathbf{u}_i + \lambda_i \partial \mathbf{Mu}_i + \partial \lambda_i \mathbf{Mu}_i , \tag{5}$$

By pre-multiplying both sides by $\mathbf{u}_i^T$ and rearranging the terms, we obtain:

$$\partial \lambda_i = \mathbf{u}_i^T \left( \partial \mathbf{K} - \lambda_i \partial \mathbf{M} \right) \mathbf{u}_i . \tag{6}$$

Now, we establish a connection between the gradient of vibration frequencies and the gradient of the mass and stiffness matrices.

### 3.3 LOSS FUNCTION FOR OPTIMIZATION

At this stage, we can optimize the material properties and geometry of the object using target eigenvalues. This optimization is performed by employing the loss function defined as:

$$L_i = ||\lambda_i^{pred} - \lambda_i^{gt}||_1, \tag{7}$$

where $\lambda_i^{gt}$ is the ground truth eigenvalue of mode $i$ and $\lambda_i^{pred}$ denotes the predicted eigenvalue.

For generality, we proceed to compute the predicted sound signal from the predicted eigenvalues as detailed in Sec. 3.3.1. Subsequently, we utilize a hybrid loss function to calculate the loss of the sound signal as detailed in Sec. 3.3.

### 3.3.1 DIFFERENTIABLE ADDITIVE SYNTHESIZER

The sound produced by a rigid-body object can be effectively modeled as a bank of damping sinusoidal oscillators. For the $i$-th mode, denoting its damping factor as $d_i$ and its amplitude as $A_i$, its frequency can be obtained by:

$$f_i = \frac{\sqrt{\lambda_i - d_i^2}}{2\pi} . \tag{8}$$

Let $h$ be the time step size, the sound signal $s_i(n)$ over discrete time steps, $n$, can be computed as:

$$s_i(n) = A_i e^{-d_i n h} \sin(2\pi f_i n h) . \tag{9}$$

Finally, the sound is produced by summing the sound signals for all modes. It's important to note that amplitudes and damping factors are designed to be learned from ground truth data, and amplitudes can implicitly include the acoustic transfer function (James, 2016). Additionally, the eigenvalues $\lambda_i$ play a crucial role in connecting the sound signal to the physical properties of the object. The computations defined in Equations 9 and 8 are evaluated in parallel along both the time and mode dimensions using PyTorch, enabling automatic differentiation.

When dealing with naturally recorded sounds that contain noise, we enhance the output of the additive synthesizer by combining it with noise filtered by an LTV-FIR filter (Engel et al., 2020). The parameters of this filter are also learnable, enabling it to adapt to real-world noise characteristics.

### 3.3.2 HYBRID LOSS FUNCTION

As suggested in previous differential audio synthesizers (Engel et al., 2020; Clarke et al., 2021), a multi-scale spectral loss is effective for measuring the difference between two audios. Given the ground-truth and predicted sound signals, we compute their spectrogram $S_i$ and $\hat{S}_i$, respectively, using a specified FFT size $i$. The loss is then defined as the sum of the L1 difference between $S_i$ and $\hat{S}_i$, as well as the L1 difference between their respective log spectrograms:

$$L_i = ||S_i - \hat{S}_i||_1 + ||\log S_i - \log \hat{S}_i||_1 . \tag{10}$$

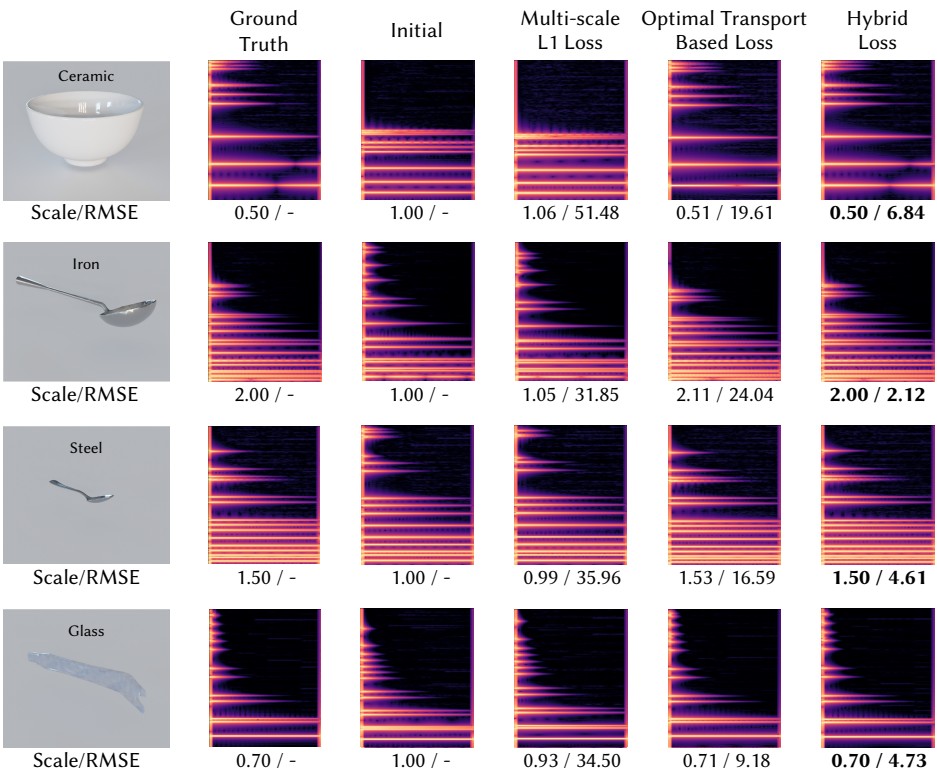

Figure 3: Ablation study on loss functions. We present the spectrograms, scaling factor, and RMSE with different setups. Across all setups, our hybrid loss function consistently outperforms, while the single multi-scale L1 loss or optimal transport-based loss shows limited effectiveness.

The total reconstruction loss is the sum of all the spectral losses with different FFT sizes, which provide varying frequency and temporal resolutions.

Traditional L1 or L2 loss can result in difficult convergence when the initial and ground truth object locations or frequencies significantly differ Xing et al. (2022). This issue also arises in differentiable sound rendering. For instance, if the initial frequency far deviates from the ground truth frequency, there may be no overlapping pixels in the spectrogram between the initial mode and target mode, causing the L1 or L2 loss to yield zero gradients and potentially leading to undesired local minima.

To address this issue, we first treat the spectrogram pixel in each frequency bin as a high-dimensional point. To measure the distance between the ground truth and predicted spectrograms, we utilize the optimal transport (Wasserstein) distance. This distance metric considers the cost of moving mass from one distribution to another. In our context, we define the unit moving cost from one frequency bin to another as their corresponding point distance. For efficiency, we employ an efficient algorithm for approximating optimal transport distances using Sinkhorn divergences (Feydy et al., 2019).

As the optimal transport-based loss tends to be less effective when the initial and target spectrograms are already well-aligned, we initially use it to achieve sufficient convergence. Subsequently, we switch to the multi-scale spectral loss for fine-tuned optimization.

## 4 INVERSE PROBLEMS AND EXPERIMENTS

We define three reasoning tasks and conduct corresponding experiments to showcase the power of our differentiable framework. First, we perform an ablation study on the loss function to validate our approach (Sec. 4.1). Next, we utilize our differentiable framework to reason about the physical parameters (Sec. 4.2), geometric shape (Sec. 4.3), and impact position (Sec. 4.4) of the object. Please refer to the supplementary video for the results of our experiments.

The real-world object data used in the experiments is sourced from the ObjectFolder-Real dataset (Gao et al., 2023), which contains multisensory data collected from 100 real-world household objects. The data for each object includes its high-quality 3D mesh, impact sound recordings, and the accompanying video footage for each impact.

Our DIFFSOUND differentiable framework is implemented in PyTorch and utilizes the Adam optimizer for optimization.

## 4.1 ABLATION STUDY ON LOSS FUNCTIONS

We first conduct an ablation study to validate the effectiveness of the hybrid loss function compared to either using a single multi-scale L1 loss or a single optimal transport-based loss.

We set up a simple case where the predicted eigenvalues can only be changed proportionally through a trainable scaling factor. We aim to optimize this scaling factor from an initial value of 1.0 to a predefined target value. We select four meshes from the dataset and manually set the material parameters, following the guidelines presented in (James, 2016).

As depicted in Figure 3, the results indicate that the optimal transport-based loss shows high effectiveness for optimizing from a bad initial state where the multi-scale L1 loss cannot work. Additionally, our hybrid loss function achieves the best performance compared to either single loss function in all experiments.

## 4.2 MATERIAL PARAMETERS REASONING

In this task, we aim to infer the material parameters from the impact sound of an object, assuming the object's geometric model is known. Initially, we estimate the damping curve. Subsequently, we optimize the other material parameters by minimizing the loss between the produced sound of our simulation framework and the target sound. To estimate the damping curve, we train numerous random modes to fit the target spectrogram using our differentiable additive synthesizer, following the approach outlined in (Engel et al., 2020) (see Figure 4). Degraded modes are then removed based on amplitude thresholds, and the damping coefficients of the remaining modes are interpolated to obtain the damping coefficients curve.

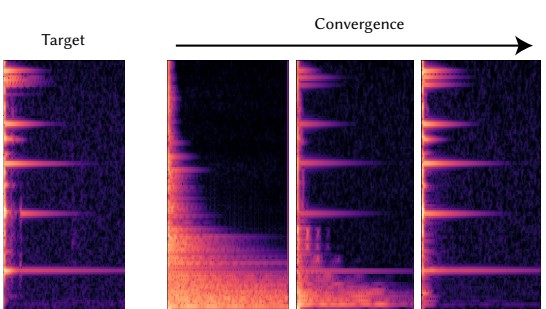

Figure 4: Training process of estimating the damping curve: We utilize 256 initial modes to comprehensively cover all target modes. After training, degraded modes are subsequently removed.

In our experiments, the material parameters include Young's modulus-to-density ratio (referred to as $\hat{E}$) and Poisson's ratio (referred to as $\nu$). Prior work (Ren et al., 2013) relied on first-order FEM and assumed a fixed Poisson's ratio, which could lead to inaccuracies. To address this limitation, we set different baselines for comparison with our method using data synthesized by second-order FEM on 16 objects. The material parameters of these objects are randomly selected from a reasonable range. Additionally, we evaluate the effectiveness of our approach using data obtained from two real-world ceramic objects.

We used relative error as a metric for $\hat{E}$, $\nu$, and sound spectrogram, defined as $l = \frac{||g-p||_2}{||g||_2}$ for ground-truth $g$ and prediction $p$. We present the quantitative results in Table 1 for synthetic data, along with qualitative examples for real-world data in Figure 5. Our DIFFSOUND demonstrates substantial improvements over all baselines across all metrics, showcasing high effectiveness even in real-world data.

|  | FEM order | Learnable $\nu$ | $\hat{E}$ Err. | $\nu$ Err. | Spec. Err. |
|---|---|---|---|---|---|
| baseline 1 | 1 | ✗ | 0.51 | 0.68 | 26.43 |
| baseline 2 | 2 | ✗ | 0.10 | 0.68 | 11.21 |
| baseline 3 | 1 | ✓ | 0.51 | 0.66 | 27.00 |
| DIFFSOUND | 2 | ✓ | **0.07** | **0.26** | **7.95** |

Table 1: Material parameter reasoning using synthetic data. Our method outperforms all baselines in terms of relative errors.

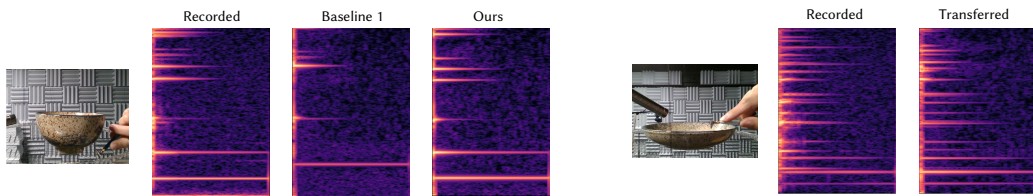

Figure 5: (Left) Material estimation from real-world recorded sound with our DIFFSOUND method and the basic baseline. (Right) Transfer of the material parameters optimized from a ceramic bowl to a plate with the same material, with additional fine-tuning of the noise filter and mode amplitude.

### 4.3 SHAPE GEOMETRY REASONING

In differentiable rendering, shape geometry reasoning is generally stable, as a few rendered images can largely determine the shape. However, determining the shape from sound is challenging because different shapes can produce similar sounds upon impact (Kac, 1966). Therefore, to improve shape reasoning, we have fixed material coefficients and imposed additional geometry constraints to ensure a stable optimization process.

In this task, we infer the shape geometry from the eigenvalues of vibration modes, which are directly related to frequencies (Eq. 8). Additionally, we constrain the tetrahedral mesh during optimization using a coarse voxel grid. Specifically, we query the SDF values from the MLP and ensure that the SDF of grid points inside the mesh is negative, while those outside are positive. This is enforced using a loss defined as the sum of absolute SDF values of those points whose SDF sign differs from the expected sign. The loss for sound constraint is defined as the L1 loss between the ground truth eigenvalues and the predicted eigenvalues of the first $k$ modes, divided by the norm of the ground truth.

In our experiments, we generate synthetic data for three objects from (Crane et al., 2013) applying a ceramic material parameter. We sample a grid of $16^3$ points within the bounding box of the mesh and choose the mode number $k$ to be 16, 32, and 64. The resolution of background tetrahedral mesh grid is $32^3$. We conduct separate experiments for each object and mode number. The geometric shape can be successfully recovered from impact sound, as illustrated by the quantitative results in Figure 6. Our approach demonstrates its ability to restore geometric features, particularly sharp detail, from sound data. This capability compensates for the loss of such details in the initial coarse mesh. The high accuracy of our approach can also be validated in the accompanying demo video, closely aligning with the ground truth.

### 4.4 IMPACT POSITION REASONING

Impact position is not explicitly optimized as a learnable parameter. However, the learnable mode amplitude $A$ in Equation 9 implicitly encodes information about the impact position.

In this task, we aim to infer the impact position from the recorded sound, given that the object's mesh is known. First, we optimize the material parameters from sound following the process outlined in Sec. 4.2. Simultaneously, we optimize the amplitudes of all modes, denoted as $\mathbf{A} = [A_0, A_1, ..., A_n]$. Then, using the estimated material parameters, we apply forward modal sound simulation, which includes acoustic transfer (Jin et al., 2022), to obtain the simulated ampli-

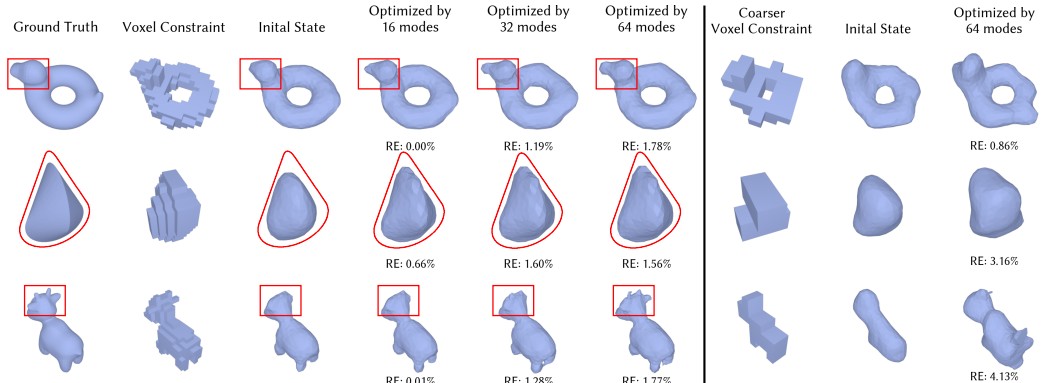

Figure 6: Optimizing shape geometry constraint through sound modes and a coarse voxel grid. We present the geometry mesh along with the relative error (RE) of eigenvalues. Our DIFFSOUND method demonstrates its capability to restore shape details from sound modes. The last three columns show results of using coarser voxel constraints, which makes the problem ill-posed. In such cases, multiple plausible shapes can produce the same sound, potentially resulting in unconventional shapes with eigenvalues closely resembling those of the ground truth.

tudes of all modes $\hat{\mathbf{A}}_i$ when impacting each mesh vertex $v_i$. We use the similarity between $\mathbf{A}$ and $\hat{\mathbf{A}}_i$ to measure the likelihood that the impact position corresponding to the recorded sound is near vertex $v_i$.

In our experiments, we choose recorded real data of a ceramic bowl from ObjectFolder (Gao et al., 2022) for our test. We use cosine similarity to measure the likelihood and compute the surface likelihood distribution, as visualized in Figure 7. Our method predicts a high likelihood around the ground truth impact position.

## 5 CONCLUSION

We have presented a differentiable sound simulator that enables inverse reasoning by computing the gradient of the simulation function with respect to input physical parameters (e.g., material parameters). We have verified the effectiveness of our loss strategy with ablation experiments and demonstrated the generality and diversity of DIFFSOUND in three application scenarios: material estimation, impact position estimation, and shape estimation. This advancement holds the potential to propel the fields of robotics and embodied AI.

Nonetheless, our framework currently faces challenges in handling complex shapes, particularly thin shells, and may not accurately model heavily nonlinear sounds. Additionally, opti-

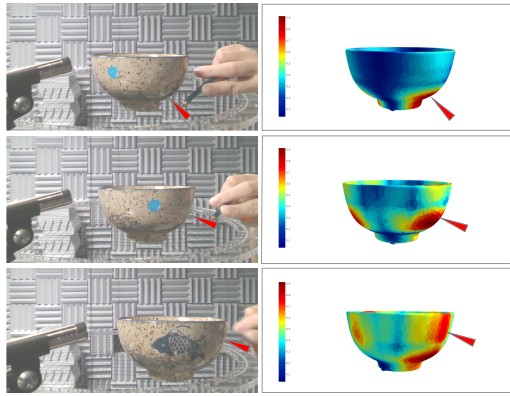

Figure 7: Visualization of the surface likelihood distribution of the impact position on the object's surface for an example object.

mizing the rendering speed to support real-time applications remains a priority. In future endeavors, we envision the development of a more comprehensive and efficient differentiable sound simulation framework, building upon the foundation laid by our DIFFSOUND.

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
