# OpenReview forum: "DiffSound: Differentiable Modal Sound Simulation for Inverse Reasoning"
_ICLR.cc/2024/Conference — Submitted to ICLR 2024_

### Official Review · Reviewer_oyHm · 2023-10-25

**Soundness:** 2 fair
**Presentation:** 3 good
**Contribution:** 2 fair
**Rating:** 3
**Confidence:** 3

**Summary:**

This paper presents an end-to-end framework for inferring the geometry and material properties of objects based on the frequency domain representation of the sound that they make.  To overcome some of the challenges, e.g., a sparse spectrogram representation, the authors propose a hybrid loss that first uses optimal transport to compute an approximate solution, and then the L1 loss to refine the solution.  Experiments are run to test material, geometry, impact positioning, independently.

**Strengths:**

+ The paper is tackling an important and challenging problem, which is especially of interest with increasing interest in AR/VR applications.
+ The writing is good, and the problem and solution are easy to follow, even for a non-expert.

**Weaknesses:**

- I accept that the problem being tackled here is challenging, but the experimentation seems very limited.  The paper more shows anecdotal examples rather than present summary statistics for a larger test set with examples for illustration.
- I am wondering how does the method fare in terms of accuracy for different materials?  Different object sizes?  and so on.

**Questions:**

How sensitive is the approach in terms of placement of the sensor?  If the microphone is too far away, does environmental effects influence the results (e.g., reverberation or other material properties that might affect reflectance, etc.?).

How much does the complexity of the shape influence reconstruction?  For example, an intricate and non-convex shape that impedes direct path to the sensor?

For Equation (10), why are both terms required?  One is just a compressed form of the other?

---

> ### Author Response · Authors · 2023-11-20
>
> 1. Indeed, if the microphone is positioned too far, the model may be susceptible to noise interference. However, it's noteworthy that reverberation or reflectance should not impact the results since our model exclusively relies on frequency information in the sound, and these phenomena do not alter the frequency of modes.
> 2. In the context of shape reconstruction, we specifically leverage the eigenmodes of objects, making the sensor placement immaterial to the results. Notably, as objects of different shapes can produce remarkably similar sounds, a challenge arises when numerous shapes yield similar sounds to the ground truth, potentially leading to optimization convergence towards an incorrect shape.
> 3. The absence of logarithmic transformation results in rapid signal decay, causing the model to focus solely on the initial spectrogram frames. The use of log spectrograms proves beneficial in capturing damping effects over time. However, it is essential to note that for modes with substantial damping factors, relying solely on log spectrograms might overlook their contribution. Hence, a combination of both approaches is preferable.

---

> > ### Comment · Reviewer_oyHm · 2023-11-23
> > **Comment from Reviewer oyHm**
> >
> > Thanks for providing the clarifications -- these make sense.

---

### Official Review · Reviewer_Kc8Y · 2023-10-30

**Soundness:** 3 good
**Presentation:** 2 fair
**Contribution:** 2 fair
**Rating:** 5
**Confidence:** 3

**Summary:**

This paper presents DiffSound framework that connects material parameters of a solid body and acoustic features from the body in a differentiable manner.
Using this model, we can construct a neural network that simulates audio signals when impacting the object or inferres the object shape from the audtory information.

**Strengths:**

The differentiable simulation is carefully derived from relevant literatures such as tetrahedral mesh, generalized eigenvalue decomposition and superimposed sinusoidal signals.

**Weaknesses:**

My major concern is that the reviewer is not convinced with the importance of shape geometry reasoning from audio signals.
I think audio modality is not as informative to recover the shape of objects. Indeed, Figure 6 gives smoothed mesh surface. The shape may be distorted without sufficient voxel constraints.

Is there any application scenario?
Perhaps this model may be applied to non-invasive examination of solid structures like impacting the surface and observing the responding signals.
However, we cannot see such usages from the current set of experimental results.

**Questions:**

1. Eq. (5) to (6)

I could follow the derivation of Eq. (6) from (5). Do you use $\partial {\bf u}_i=0$ or some transformation of ${\bf u}_i ^T {\bf M} {\bf u}_i$?

2. What do you mean by *hybrid loss*?

Is it hybrid because linear and logarithm error is combined as in Eq. (10)?
Or does this mean the use of $\ell_1$ loss and OT-based loss?

3. Using ground truth $\nu$

In the result of Table 1, how is estimation of $\nu$ critical to reduce the error in the spectrogram?
Does the error reduce if ground truth Poisson's ratio $\nu$ is given?

---

> ### Author Response · Authors · 2023-11-20
>
> 1. In the general eigendecomposition definition, $u_i^TMu_i = I$.
> 2. The term "hybrid" denotes the utilization of both $\ell_1$ loss and OT-based loss.
> 3. Indeed, if the ground truth vector $\nu$ is provided, the error will diminish. Conversely, if $\nu$ is poorly estimated, the error will be substantial.

---

### Official Review · Reviewer_aeBV · 2023-10-31

**Soundness:** 2 fair
**Presentation:** 2 fair
**Contribution:** 2 fair
**Rating:** 5
**Confidence:** 4

**Summary:**

The paper proposes a differentiable sound simulation framework called DIFFSOUND, containing three components. The first component is a differentiable tetrahedral representation, which uses implicit neural representation to encode SDF values and convert the encoded SDF into an explicit tetrahedral mesh. The second component uses a high-order finite element method to optimize material properties and shape parameters. In the end, an additive audio synthesizer synthesizes the sound.

**Strengths:**

1) The idea of building a differentiable sound simulation pipeline is very interesting. While the task is challenging, I am glad that the authors come up with a solution that will definitely be useful for various applications.
2) The component introduced in this work is highly interpretable. To my best knowledge, physical properties such as Young's modulus and Poisson's ratio were not modeled in the previous audio synthesizers.
3) Three inverse problems are conducted, and the results look reasonable.
4) The supplementary includes the code, which will be useful for reproduction.

**Weaknesses:**

1) One main concern is that the paper writing is very rough. For example, in 3.1 Differentiable tetrahedral representation, there is no formal mathematical definition for the input-output, INR, tetrahedron mesh, and transformation function. The description in 3.1 is high-level and not informative. In 3.3, the loss equations 7 and 10 use the same annotation, but the $i$ means totally different things. Section 4 is a weird combination of both ablation studies and experiments on three inverse problems. I believe the inverse problems should take an independent section because it is one of the main contributions of this work. In tables and figures, annotations like baselines 1, 2, and 3 could be confusing since there is no corresponding description in captions. While each of these items could be a minor issue, the overall reading experience is actually bad.
2) I am interested in how fast the optimization could be done for each object, but there is no clue in the paper. While it is okay that the current approach could not support real-time applications, it should contain an analysis for the optimization time.
3) One thing that confuses me is the ground truth eigenvalues. How do you obtain the ground truth eigenvalue? Without supervision on the eigenvalues, the optimization problem becomes much more challenging. Is it possible to optimize with only the audio loss?
4) From my own experiences, the Wasserstein distance is indeed helpful in bridging the ground truth and predicted spectrograms. However, the switch timing between Wasserstein loss and L1/L2 spectrogram loss is undefined. How do you determine 'sufficient convergence'?

**Questions:**

See my questions in the weakness section.

---

> ### Author Response · Authors · 2023-11-20
>
> 1. The experiments concerning shape geometry reasoning are conducted on a synthetic dataset, where ground truth eigenvalues are computed using the standard modal analysis process. For material parameters reasoning, real-world datasets are employed, relying solely on audio loss for optimization.
> 2. Currently, we arbitrarily define 1000 epochs as a point of 'sufficient convergence.'

---

### Official Review · Reviewer_cDNR · 2023-11-10

**Soundness:** 2 fair
**Presentation:** 3 good
**Contribution:** 2 fair
**Rating:** 3
**Confidence:** 3

**Summary:**

The paper describes a differential simulation framework for sound synthesis of physical objects impacts. The framework is a pipeline that employs a NeRF-like MLP to reconstruct the Signed Distance Function and translate it into the shape of the object. These are then being used by Finite Elements Method to recover object shape and an Additive Synthesizer to generate sound which is optimized by minimizing loss between the expected and groundtruth spectrograms. Experiments are performed on ObjectFolder-Real dataset for sounds from 100 objects.

**Strengths:**

1. The work proposes an additional step of recovery of Signed Distance Function done by MLP to assist with object shape recovery and synthesis of impact sound of the object.

2. Synthesized results appear to be corresponding to objects and their expected sounds.

3. The paper is well written.

**Weaknesses:**

1. The choice of baselines and whether these are strongest possible baselines is unclear.

2. The experiments are done on 100 objects only.

3. Train/validation/test split is not specified and thorough quantitive accuracy of these is not presented.

4. Technical contribution is limited since the components of the pipeline are standard. Ablations with extensions of the components are needed to examine whether these are optimal for sound synthesis.

**Questions:**

1. The current pipeline is split between a neural network approach and FEM simulator. Could both steps be modeled with a neural networks?

2. How would the work compare with impact sound generation from videos through diffusion model?
Su, Kun, et al. "Physics-Driven Diffusion Models for Impact Sound Synthesis from Videos." Proceedings of the IEEE/CVF Conference on Computer Vision and Pattern Recognition. 2023.

3. What is the computational complexity of the pipeline?

---

> ### Author Response · Authors · 2023-11-20
>
> 1. While it is possible to model both steps using neural networks, achieving comparable accuracy to a classic solver is challenging.
> 2. Our paper specifically addresses the inverse problem of inferring physical parameters from impact sound. In contrast, the referenced paper by Su, Kun, et al. focuses on predicting sound from visual features.
> 3. The computational complexity of the pipeline can be approximated as O(MN^2), where N represents the number of vertices and M is the number of modes. This estimation accounts for the most time-consuming aspect, which is the eigen decomposition process.

---

### Meta-Review · Area_Chair_BsrA · 2023-12-06

**Metareview:**

The submission simulates and estimates the sound properties of objects. It has received 4 negative reviews, which were critical of the following aspects of the paper:

- Weak experiments on a small dataset,
- Lack of details of the experimental setup,
- Choice of baselines,
- Limited novelty,
- lacking clarity in presentation and writing.

The authors provided some answers, but only extremely sparingly. Some easy questions have been ignored, as for instance the train/validation/test split. In general, the answers were not sufficient.

The AC agrees with the reviewers and judges that the paper is not suitable for publication at ICLR 2024.

**Justification For Why Not Higher Score:**

-

**Justification For Why Not Lower Score:**

-

---

### Decision · Program_Chairs · 2024-01-16

Reject